# Neuroticism and Psychological Stress in Patients Suffering Oral Lichen Planus: Research Evidence of Psycho-Stomatology?

**DOI:** 10.3390/healthcare11121738

**Published:** 2023-06-13

**Authors:** Luis Alberto Gaitán-Cepeda, Diana Ivette Rivera-Reza, María del Carmen Villanueva-Vilchis

**Affiliations:** 1Department of Oral and Maxillofacial Medicine and Pathology, Research and Graduate Division, Dental School, National Autonomous University of Mexico, Mexico City 04510, Mexico; dianrvr@fo.odonto.unam.mx; 2Department of Oral Public Health, School of Superior Studies, National Autonomous University of Mexico, León 37684, Mexico; cvillanueva@enes.unam.mx

**Keywords:** oral lichen planus, psychosocial factors, psychological stress, neuroticism, oral health-related quality of life, psycho-stomatology

## Abstract

Psychosocial factors influence the development, exacerbation, or aggravation of some oral diseases. However, the possible relationship between personality traits, affective disorders, and psychological stress in oral diseases, and their impact on oral health-related quality of life (OHRQoL), has not been fully clarified. The aim of the present study was to determine the association of neuroticism and stress with the presence of oral lichen planus (OLP), and to discover whether or not these factors impact OHRQoL. This is a case-control study matched for age and sex. The case group (OLP group) was composed of 20 patients diagnosed with OLP, while 20 people with a diagnosis of lesions not associated with stress formed the control group. Three instruments were used: the Holmes–Rahe Social Readjustment Scale, the Five Factor Personality Model, and the OHIP-49. Neuroticism obtained a score of 25.5 (±5.4) in the OLP group, which was higher than the control group value (21.7) (±5.1) (*p* = 0.03). The OLP group showed a worse quality of life (*p* < 0.05); the most affected dimensions were psychological discomfort and physical disability. It is important to include a psychological profile to establish a comprehensive treatment for these patients. We propose the recognition of a new area of clinical oral medicine: psycho-stomatology.

## 1. Introduction

The development, exacerbation or aggravation of some oral mucosa diseases are profoundly influenced by psychosocial factors, personality traits, affective disorders, and psychological stress [1,2,3]. Personality traits refer to consistencies in behavior and describe individual personality differences. The most widely accepted trait model is the Five Factor Model (FFM) [4]. The FFM proposes the following factors: 1. Energy or extroversion, characterized by a predisposition to positive emotions and traits related to activity and energy, with a tendency to be sociable, active, talkative, optimistic, and assertive; 2. Emotional stability or neuroticism is characterized by individuals prone to tension, moodiness, anxiety, irritability, emotional instability, and psychological distress, with a tendency to chronically experience negative emotions such as depression, anxiety, and anger; 3. Openness to experience relates to scientism, divergent thinking, political liberalism, proactive pursuit, broad interests, imagination, and insight; agreeableness includes sympathy, kindness, and affection; 4. Kindness as a prosocial orientation, with a tendency to be compassionate, cooperative, friendly, considerate, generous, trusting, forgiving, and willing to compromise one’s interests with others; and 5. Conscientiousness or responsibility: a tendency toward organization, self-control, perseverance, and motivation in goal-oriented behaviors [4,5]. Each of the five personality traits impacts health-related outcomes. Psychological stress is the condition resulting from the perception of a discrepancy, real or not, between the demands of a situation and the resources available to the individual [6]. Chronic psychological stress triggers non-specific systemic biochemical responses that produce imbalances and disease, constituting the so-called General Adaptation Syndrome (GAS) [7]. Three stages are identified in the GAS: alarm reaction, resistance, and exhaustion [7]. In the resistance stage, the continuous stress will lead to neurological changes and collapse of the hormonal system, producing adaptive diseases such as peptic ulcer, hypertension, hyperthyroidism, and immune deficiencies. In the exhaustion stage, an abnormal lower parasympathetic function results in physical deterioration, or even death [7].

Stress-related oral diseases (SROD) negatively impact the quality of life of individuals. Oral health-related quality of life (OHRQoL) refers to the impact of some oral conditions and diseases on aspects of daily functioning such as eating, nutrition, social interaction, emotional and psychological functions, and financial repercussions [8]. The most studied SRODs are oral lichen planus (OLP), recurrent aphthous stomatitis (RAS), and burning mouth syndrome (BMS). OLP is a chronic immune-mediated inflammatory disease of the oral mucosa that affects 1–2% of the general adult population, mainly women between 30 and 60 years of age [2]. Although its pathogenic mechanism is not fully understood, genetic, environmental or local factors and psychological distress have been mentioned. Of particular interest to this study, exacerbation of OLP is related to periods of psychological stress and anxiety [9,10]. Clinically, OLP is characterized by different types of lesions, white striations (Wickham’s striae), white papules or plaques, usually symmetrical and bilateral; erythema, erosions, or blisters predominantly affecting the buccal mucosa, tongue, and gingivae [11,12] (Figure 1). The histopathologic features of OLP include hyperkeratosis (reticular OLP), epithelial atrophy or loss of the epithelial layer with the presence of an ulcer (erosive OLP), liquefaction degeneration of the basal layer of epithelium, apoptosis of the keratinocytes (Civatte bodies), saw-tooth epithelial ridges, the presence of a homogeneous eosinophilic deposit at the basal membrane, and dense subepithelial lymphocytic infiltrate [13]. Symptoms include sensitivity of the oral mucosa to hot or spicy foods and painful oral mucosa. OLP has periods of relapses and remissions, and exacerbations [11,14,15]; the last one is related to periods of psychological stress and anxiety [9,10]. Erosive OLP is considered a potentially malignant oral mucosa disorder, with a risk of progression to cancer of 0–10% [16,17]. A systematic review and meta-analysis reported malignant transformation rates of 1.14%. Topographically, the tongue carried on a significantly higher risk of malignant transformation, and was the most frequent area affected by cancer in OLP. Consequently, atrophic-erosive lesions on the tongue are of particular concern [17]. The malignancy mechanism of OLP is still unknown. However, a factor proposed in the pathogenesis of OLP are cancer stem cells, which are increasing in atrophic-erosive clinical forms compared to reticular ones [18]. To date, there is no curative treatment for OLP. Thus, the primary objective of its treatment is the control of clinical episodes. The proposed treatments are numerous and varied, where topical glucocorticoids in adhesive vehicles (orabase) or in mouthwashes [11,14,15] is considered to be the first choice of treatment. However, there are cases that are refractory to treatment, or with worsening of clinical lesions, that are treated with systemic corticosteroids. Much less frequently, immunomodulatory therapy (azathioprine, cyclosporine, and methotrexate) and biologic agents (afalizumab, alefacept) have been used [15].

The relationship between psychosocial and personality trait factors with SRODs has been studied. However, in most of the studies, these variables have been reviewed independently. There exists little information, to our knowledge, about the interrelation in the same patient of several psychosocial factors in the development of SROD. Therefore, the main objective of the present study was to determine whether there was an association between personality traits and stress levels in a group of patients diagnosed with OLP; furthermore, whether these psychological factors impact the OHRQoL, precisely on the dimensions associated with psychological adjustment (psychological discomfort, psychological disability, and social disability).

## 2. Materials and Methods

A case-control study (one control for each case) was performed in the Oral Medicine Clinic, Division of Graduate Studies and Research of Dental School, National Autonomous University of Mexico by non-probabilistic sampling, from August 2018 to December 2019. The research protocol was approved by the Research and Ethics Committee of the Dental School of the same institution (Register C.I.E.: 0610/11/2018).

All patients attending for the first time, and without treatment at least six months before their inclusion in the study, were invited to participate. The case group consisted of patients with a clinical diagnosis of Oral Lichen Planus and diagnostic confirmation by histopathological study. The control group consisted of patients with benign oral mucosal, who were matched for age and gender with the case group. Individuals with a diagnosis of premalignant lesion or neoplasia (benign or malignant) were excluded from both groups, as well as those with any condition that prevented the correct response to the instruments applied during the interview. The clinical diagnosis was established, independently of the objectives of this study, by two experts in oral medicine (LAGC, DIRR). Before the application of the surveys, each participant was given a brief explanation of the scope and objectives of the study, and those who agreed to participate signed an informed consent form, which guaranteed confidentiality in the handling of the data and the possibility of withdrawing from the study at any time.

Only one interviewer (DIRR) applied three instruments to both study groups (cases and control) through an interview. The stress level was determined through the Holmes–Rahe Social Readjustment Scale [19]. This 43-item instrument measures the magnitude of stress a person has experienced during the past year. Each item has a specific value; the total value determines the degree of stress manifested in the last year. It is categorized as low stress (less than 150), medium stress (151 to 300), and high stress (301 or more). According to their stress levels, the patients in both study groups were divided into two subgroups: patients with high stress, and patients without high stress; the latter consisted of people with medium and low stress. The Big Five Personality Factor Model (Big Five Inventory) [20] was used to determine personality traits. This tool is a questionnaire based on the five personality factors: (1) Neuroticism, (2) Responsibility, (3) Agreeableness, (4) Extroversion, and (5) Openness to experience. It consists of 44 items. Each item is a short sentence that the respondent evaluates through a 5-point Likert scale (1. strongly disagree; 2. disagree; 3. neither disagree nor agree; 4. agree; and 5. strongly agree) of how much it describes or fixes them. The highest sum determines the predominant personality trait (Expressiveness). To establish the impact on quality of life, the Mexican version of the OHIP-49 instrument was used [21]. OHIP is a tool whose main objective is to quantify the frequency with which patients face difficulties carrying out daily activities because of oral conditions or diseases. OHIP-49-mx has 49 items representing seven dimensions: (1) functional limitation, (2) pain, (3) psychological discomfort, (4) physical disability, (5) psychological disability, (6) social disability, and (7) disability (incapacity). The responses were recorded on a Likert-type scale, whose values ranged from 0 (never) to 4 (always). Zero is the minimum possible score, while 196 is the maximum, calculated using the additive response method. The higher values, the worse quality of life. The 50 percentile (median) was used as a cut-off point to determine the negative impact on quality of life. Thus, those values greater than the 50 percentile scores (median) were considered to reflect poor quality of life.

At the beginning of the interview, the interviewer collected sociodemographic information, age, gender, marital status, and occupation. With the obtained data, an ex profeso database was built. The STATA 14.1^®^ statistical package was used for statistical analysis, and descriptive analysis was obtained for the sociodemographic variables. Normality was assessed using the Shapiro–Wilk test. Bivariate analysis was carried out using t-Student, chi2, and Fisher exact; finally, a multivariate logistic model was undertaken.

## 3. Results

Forty participants were included, consisting of 20 cases and 20 controls. The mean age of the control group was 57.4 (±15.37 years), and the case group was 58.05 (±15.61 years). In both groups, 15 were female and five were male. The most frequent marital status in the OLP group was “single” with 7 cases (35%), followed by “married” with five cases (25%). In the control group, 10 individuals (50%) were “married” and 5 “single” (20%). The most frequent occupation in the OLP group was “home” and “employee” with 30% each, followed by “pensioner” with 20%. In the control group, 50% of the people were engaged in “home”, followed by “employed” and “retired”, with 15% each. None of these demographic variables presented a statistically significant difference between the groups (Table 1). The most frequent diagnoses in the control group were “prosthetic stomatitis” with 6 (30%), “no lesion” with 3 individuals (15%), and “fibrous hyperplasia” in 2 (10%) of patients.

### 3.1. Psychosocial Variables

Regarding personality traits, the OLP group showed the highest scores in Openness (34.2; ±4.7), Responsibility (32.2 ± 4.5), and Agreeableness (30.5 ± 3.6); while in the control group, the highest personality scores were Openness (38.2 ± 4.1), Agreeableness (34 ± 4.3), and Responsibility (34.3 ± 3.6). Although there were similarities, there was one important difference between the scores of both groups, which was the trait of Openness, where the control group obtained a mean of 38.2 (±4.1) compared to 34.2 (±4.7) for the OLP group (*p =* 0.01).

The study variable, Neuroticism, obtained a score of 25.5 (±5.4) in the OLP group, and for the control group the value obtained was 21.7 (±5.1), lower than that of the OLP group; this difference was statistically significant (*p* = 0.03). For the stress variable, 45% of the patients in the OLP group showed high stress, and in the control group, 45% of the subjects showed medium stress; however, these differences did not show a statistical significance (Table 1). When only the high stress patients were analyzed, the OLP group was found to have the highest values; however, these differences were not statistically significant (*p* > 0.05) (Table 1).

### 3.2. Oral Health-Related Quality of Life

The total OHIP value in the OLP group was 78.8 (±39). The dimensions with the highest scores were those associated with functionality; that is, functional limitation (18.3 ± 7.2), pain (15.7 ± 8), and physical disability (14.5 ± 7.9). On the other hand, the control group had a total OHIP value of 52.7 (±44.8), significantly lower than that obtained in OLP subjects (*p =* 0.056). Similarly, the dimensions with the highest scores were functional limitation (13.4 ± 9.9), pain (11.6 ± 9.9), and physical disability (8.6 ± 9.9). It is necessary to mention that the OLP group presented higher values in all the OHIP-49 dimensions than the control group, being the difference statistically significant in the physical disability dimension (*p =* 0.042) and the total of the instrument (*p =* 0.056) (Table 1). The physical disability dimension in the OLP group was 68.6% higher than in the control group. The greatest relative difference was observed in the dimensions of disability, with an increase of +112.5%, social disability with a +85% increase, and physical disability with a +68.6% increase.

When establishing a good or bad quality of life according to the 50th percentile (median) as a cut-off point, the OLP group presented a greater amount of people with a poor quality of life in all dimensions than the control group, with a statistically significant association in the dimensions of psychological discomfort and physical disability (Table 2). According to the results obtained, suffering OLP confers a four-fold possibility of having a poor quality of life related to psychological discomfort (*p =* 0.01), and a three-fold probability of having a poor quality of life related to physical disability (*p =* 0.03).

When neuroticism was related to OHRQoL in the OLP group, it was higher in those with both a good and poor quality of life, although these associations were not statistically significant (Table 1).

The marital status variable was included in the model. However, no associations were found between the personality traits Openness, Neuroticism, and Agreeableness (Table 3), with stress level and marital status. The logistic regression models included the dimensions of psychological discomfort and physical disability, since statistically significant associations were observed in these dimensions. The results (Table 3) show that having OLP increased by 37 times the likelihood of having a poor quality of life related to the psychological discomfort dimension (*p =* 0.03), as well as 0.47 times the probability of the presence of neuroticism (*p =* 0.04). For the physical disability dimension, the results are shown in Table 3, in which suffering from OLP increases up to 44 times the possibility of having a poor quality of life related to that dimension (*p =* 0.028).

## 4. Discussion

This study aimed to determine the possible association between stress, personality traits, and SROD, and whether the OHRQoL is impacted by the dimensions of psychological disability and social disability. Our results showed that patients suffering from OLP are predominantly subjects with higher neuroticism, high stress, less Openness and Agreeableness, and poor quality of life, specifically in relation to physical disability and social disability.

A well-established association exists between psychological stresses and some oral mucosa diseases. In this way, a synergic association between OLP, anxiety, and depression exists, resulting in a poor quality of life and increased stress levels. Anxiety is probably the most influential covariate associated with OLP. Patients with OLP present a risk of suffering from depression, anxiety, and stress significantly higher than the general population [22]. Exposure to stress triggers or exacerbates the clinical events of these diseases. Čanković et al. conclude that the degree, number, and type of stressful changes experienced play an essential role in the onset or reactivation of OLP [23]. Interestingly, and despite OLP patients showing a higher frequency of depression than the general population [9], the majority of OLP patients are unaware that they are suffering from anxiety or depression; consequently, they had never had a psychiatric evaluation [16]. Our data showed slightly more patients with high stress in the case group.

Čanković et al. showed that patients suffering from OLP tend toward neuroticism [23]. Greater neuroticism is related to higher depression levels. Regarding personality, OLP patients are strongly correlated with depression and anxiety [1,24,25,26]. On the other hand, higher levels of perceived stress were related to higher scores of OLP, directly associated with neuroticism [9]. Depression can interact with life stress through two mechanisms: (1). Those with high levels of stress may be more prone to depression, and (2). Those individuals who are already feeling somewhat depressed or anxious may feel even more helpless and hopeless when faced with greater levels of life stress [9]. The personality trait closely related to stress is neuroticism. Individuals who are highly neurotic tend to experience negative emotions such as anxiety, anger, or depression; they are also more prone to feelings of worry, inadequacy, and nervousness. Individuals with high neuroticism scores also had high perceived stress scores [9]. Our results suggest that neuroticism is related to perceived stress scores.

The development of the strong relationship between stress and neuroticism results in an increase in the amount of stress felt, and vice versa. Stressful events in an individual’s life may increase their levels of neuroticism [9]. Our results are similar, since patients suffering from OLP showed higher neuroticism predominance and lower values for openness and agreeableness. Subjects with higher neuroticism scores had higher OHIP-49 scores. Nevertheless, the nature of the mucosal condition influences the OHIP scores and the item prevalence.

Suffering from any SROD has a negative impact on OHRQoL [24,27,28,29]. Moreover, it has been reported that the most affected dimensions are pain, psychological discomfort, physical, psychological and social disability, and incapacity [24,28,29]. Our data are consisted with what has been reported, because the highest OHIP values were found in patients suffering from OLP, where the most affected dimensions were psychological discomfort and physical disability. Our result is consistent with previous reports that revealed that the “physical pain dimension” is related to neuroticism and gender, while “psychological discomfort dimension” and “psychological limitation dimension” were also linked to neuroticism and gender [10].

The clinical variety of OLP is an important variable, because the bullous/erosive type significantly influenced neuroticism and poor OHRQoL. Personality traits, especially neuroticism, influence the subjective questions about oral health-related quality of life. Neuroticism negatively impacts daily performance among patients with mucosal diseases. It has been suggested in the scientific literature that neuroticism can be considered a great predictive factor in the quality of life concerning aesthetics and psychosocial impact. Clinical events of OLP can worsen during stressful events [10].

The originality of the present study lies in the fact that it has analyzed psychosocial variables such as personality traits, psychological stress, and OHRQoL simultaneously because they are simultaneous and concomitant events, providing a comprehensive picture of patients suffering from SROD. To our knowledge, this is the first time that a comprehensive analysis of psychosocial variables and SROD has been performed. The bidirectional link between mood disorders and OLP is well recognized, and could support a bidirectional connection between the immune and central nervous systems. The central elements of psychological stress are the hypothalamus-pituitary-adrenal (HHA) axis and the sympathetic-adrenal-medullary (SSM) axis [29,30,31]. The hypothalamus integrates physical and emotional stimuli as a consequence of stress exposition, and responds by secreting corticotropin-releasing hormone (CRH) from the paraventricular nucleus. CRH, in turn, induces the release of the adrenocorticotropic hormone (ACTH) by the pituitary gland, which stimulates the production of glucocorticoids (cortisol) and androgenic hormone dehydroepiandrosterone (DHEA) from the adrenal gland. Psychological stress increases cortisol and catecholamines as an immediate anti-stress response, but will also produce immunodeficiency [30,31] and/or amplified cytokine production [22], resulting in atopic autoimmune disease [30,32]. Severe depression and stress have been linked to peripheral inflammation and the exacerbation of chronic autoimmune diseases [33] showing increased blood neutrophils, monocytes, and inflammatory mediators [34,35].

This influence may occur many years before, shortly before, or during the clinical course of the disease. Similarly, OLP is considered to be an inflammatory disease of possible autoimmune origin. There is epidemiological evidence suggesting that most patients with autoimmune disorders show increased emotional stress before the onset of the clinical event. A vicious circle is established because autoimmune disorders cause stress, increasing the gravity of clinical events [36].

Our results suggest that personality traits plus stress impact the development and exacerbation of an SROD. Therefore, clinical worsening will probably be identified more intensely in people with this personality, establishing a vicious cycle, which will need to be interrupted not only by allopathic treatment application, but also psychological treatment (Figure 2). In recent decades, knowledge of the importance and influence of psychological and biological aspects of stress has been integrated into its study. It is necessary to highlight the importance of incorporating psychosocial variables in the research and understanding and diagnosing various diseases.

The incorporation of psychological evaluation is fundamental, since psychosocial stimuli and systemic physiological responses are bent together. The association has led to close and often bi-directional interrelationships, generating new areas of clinical medicine such as psycho-dermatology, psycho-neuroendocrinology, psycho-immunology, and psycho-cardiology [37]. The results obtained in the present work show that personality traits and exposure to psychological stress are related to the presence of OLP, a stress-related oral disease. We suggest incorporating psycho-stomatology in oral medicine, which will justify the need to comprehensively evaluate patients suffering from SROD, including OLP, RAS, and BMS. This particularly applies to patients with refractory to pharmacological treatment, or those with long-standing diseases.

The present study has weaknesses, including the lack of quantitative tools for the measurement of stress-associated biochemical components such as cortisol in blood or saliva, which are necessary to confirm the condition in these patients. This information will undoubtedly support the relationship between emotional (psychosocial) and adaptive or pathological physiological changes, and should be considered in future research [38,39]. On the other hand, stress reactivity is variable in each individual, which limits the generalization of the results [30]. Due to its heterogeneity, there is no single definition of stress, which makes it difficult to measure. The most widely used and validated tools for the measurement of psychological stress are patient self-reports. In addition, it should be noted that the variables of interest, stress, and personality may be variables that are highly dependent on each other. Therefore, the results should be viewed with caution until confirmed by other research groups.

## 5. Conclusions

It is necessary to incorporate screening tools in the diagnostic strategy to identify patients who require psychological care. The present study evidences the importance of taking into account the psychological state of patients at the time of their anamnesis, since people with traits of neuroticism may have less emotional stability. However, due to the small sample of patients, a large study will have to be performed to confirm our results. Therefore, it is suggested that the analysis of illnesses should not only be limited to their biological characteristics but also include psychosocial aspects, bearing in mind that improving the psychological status of OLP patients could reduce the symptoms, increase treatment adherence, and improve prognosis.

## Figures and Tables

**Figure 1 healthcare-11-01738-f001:**
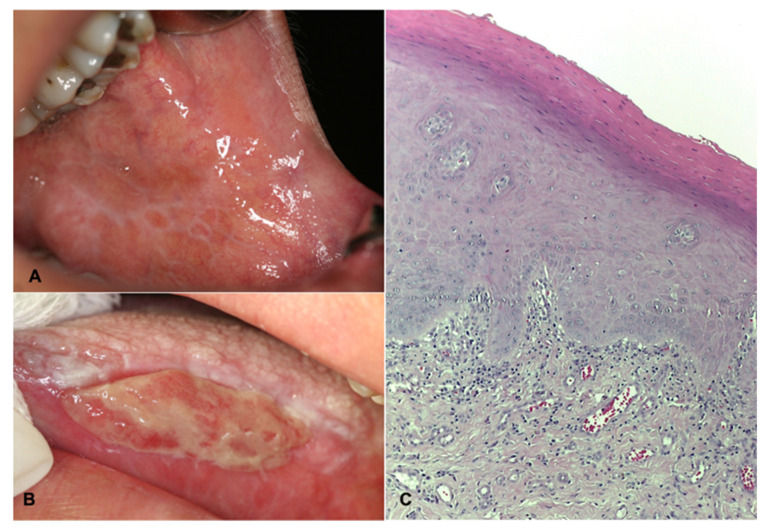
Clinical and histopathologic features of OLP. (**A**) Reticular variant where Wickham’s striae are identified in the buccal mucosa. Female, 45 years old. (**B**) Bullous variant, the ulcer covered by fibrin, is identified, surrounded by an erythematous halo and Wickham’s striae. Female, 68 years-old. (**C**) Photomicrograph at 100× H&E showing hyperkeratosis, liquefactive degeneration of the stratum basale epithelium, and chronic inflammatory infiltrate arranged in a subepithelial band.

**Figure 2 healthcare-11-01738-f002:**
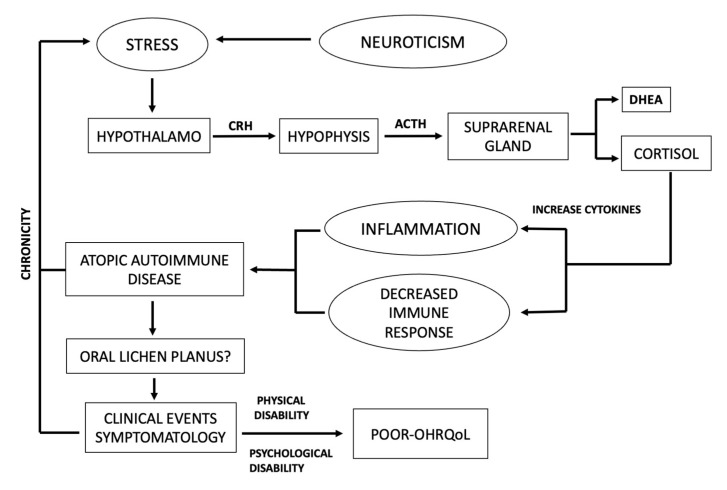
Schematic representation of influence of psychological stress to development and aggravation of autoimmune disease.

**Table 1 healthcare-11-01738-t001:** Sociodemographic characteristics, personality traits and stress levels of cases (OLP) and controls.

Sociodemographic Characteristics		Cases (*n* = 20)	Controls (*n* = 20)	*p*
**Age**		58.05 ± 15.61	57.4 ± 15.37	0.89 ^§^
**Gender**	Female	15 (75%)	15 (75%)	
	Male	5 (25%)	5 (25%)	1.00 *
**Marital status**	Single	7 (35.0%)	4 (20.0%)	
	Married	5 (25.0%)	10 (50.0%)	
	Divorced	4 (20.0%)	3 (15.0%)	
	Widowed	4 (20.0%)	3 (15.0%)	0.43 *
**Occupation**	Home	6 (30.0%)	10 (50.0%)	
	Merchant	2 (10.0%)	3 (15.0%)	
	Employee	6 (30.0%)	2 (10.0%)	
	Pensioner/retired	4 (20.0%)	3 (15.0%)	
	Student	2 (10.0%)	2 (10.0%)	0.50 *
**Personality categories**				
Openness (R: 1–50)		34.2 (±4.7)	38.2 (±4.1)	0.01 ^§^
Neuroticism (R: 1–40)		25.5 (±5.4)	21.7 (±5.1)	0.03 ^§^
Extroversion (R: 1–40)		24.5 (±4.6)	25.6 (±2.7)	0.36 ^§^
Responsibility (R: 1–45)		32.2 (±4.5)	34.3 (±3.6)	0.10 ^§^
Agreeableness (R:1–45)		30.5 (±3.6)	34 (±4.3)	0.01 ^§^
**Stress categories**				
High (R: 301–max)		9 (45.0%)	8 (40.0%)	
Medium (R: 151–300)		5 (25.0%)	9 (45.0%)	
Low (R: 0–150)		6 (30.0%)	3 (15.0%)	0.37 *
Total, obtained		316.1 (±257.0)	352.4 (±251.5)	0.65 ^§^
**OHIP-49**				
Functional limitation (R: 0–36)		18.3 (±7.2)	13.4 (±9.9)	0.082 ^§^
Physical pain (R: 0–36)		15.7 (±8.0)	11.6 (±9.9)	0.158 ^§^
Psychological discomfort (R: 0–20)		11.2 (±5.7)	7.6 (±6.8)	0.077 ^§^
Physical disability (R: 0–36)		14.5 (±7.9)	8.6 (±9.9)	0.042 ^§^
Psychological disability (R: 0–24)		8.3 (±6.4)	6.0 (±6.5)	0.277 ^§^
Social disability (R: 0–20)		3.7 (±4.3)	2.0 (±2.6)	0.143 ^§^
Handicap (R: 0–24)		6.8 (±7.1)	3.2 (±4.2)	0.060 ^§^
Total (R: 0–196)		78.8 (±39.0)	52.7 (±44.8)	0.056 ^§^
**People with high stress (*n* = 17)**				
Neuroticism		28.0 (±5.3)	24.5 (±5.8)	0.2262 ^§^
Psychological discomfort		13.5 (±4.4)	11.2 (±7.2)	0.4328 ^§^
Psychological disability		10.3 (±6.7)	9.5 (±6.2)	0.7943 ^§^
Total, OHIP		97.3 (±34.6)	76.9 (±41.8)	0.2867 ^§^
**Mean neuroticism according to quality of life**				
Good		22.3 (±5.6)	20.5 (±4.6)	0.4429 ^§^
Poor		27.4 (±4.6)	24.1 (±5.4)	0.1769 ^§^

±: standard deviation, *p*: statistical significance at 95%, R: reference value, ^§^: *t*-Student, *: Fisher’s exact chi2 direct source.

**Table 2 healthcare-11-01738-t002:** Distribution of good and poor quality of life in cases and controls.

Quality of Life	Cases		Controls		Total		OR	CI	*p*
	*n*	%	*n*	%	*n*	%			
** *Functional limitation* **									
Good	9	*45*	13	*65*	22	*55*	2.27	0.54–9.82	0.20
Poor	11	*55*	7	*35*	18	*45*			
**TOTAL**	**20**	** *100* **	**20**	** *100* **	**40**	** *100* **			
** *Physical pain* **									
Good	8	*40*	12	*60*	20	*50*	2.25	0.54–9.61	0.21
Poor	12	*60*	8	*40*	20	*50*			
**TOTAL**	**20**	** *100* **	**20**	** *100* **	**40**	** *100* **			
** *Psychological discomfort* **									
Good	6	*30*	14	*70*	20	*50*	5.44	1.18–26.25	0.01
Poor	14	*70*	6	*30*	20	*50*			
**TOTAL**	**20**	** *100* **	**20**	** *100* **	**40**	** *100* **			
** *Physical disability* **									
Good	7	*35*	14	*70*	21	*52.5*	4.33	0.97–20.19	0.03
Poor	13	*65*	6	*30*	19	*47.5*			
**TOTAL**	**20**	** *100* **	**20**	** *100* **	**40**	** *100* **			
** *Psychological disability* **									
Good	9	*45*	12	*60*	21	*52.5*	1.83	0.44–7.72	0.34
Poor	11	*55*	8	*40*	19	*47.5*			
**TOTAL**	**20**	** *100* **	**20**	** *100* **	**40**	** *100* **			
** *Social disability* **									
Good	8	*40*	13	*65*	21	*52.5*	2.77	0.65–12.20	0.11
Poor	12	*60*	7	*35*	19	*47.5*			
**TOTAL**	**20**	** *100* **	**20**	** *100* **	**40**	** *100* **			
** *Handicap* **									
Good	9	*45*	13	*65*	22	*55*	2.27	0.54–9.82	0.20
Poor	11	*55*	7	*35*	18	*45*			
**TOTAL**	**20**	** *100* **	**20**	** *100* **	**40**	** *100* **			
** *Total, OHIP* **									
Good	7	*35*	13	*65*	20	*50*	3.45	0.79–15.46	0.06
Poor	13	*65*	7	*35*	20	*50*			
**TOTAL**	**20**	** *100* **	**20**	** *100* **	**40**	** *100* **			

*n*: number of participants, %: percentage, OR: odds ratio, CI: 95% confidence interval, *p*: statistical significance 95% chi2. Direct source.

**Table 3 healthcare-11-01738-t003:** Logistic regression model with psychosocial variables, and by dimensions. Psychological discomfort and physical disability (OHIP-49).

Model Variables	*p*	OR	CI 95%
**Marital status**			
Married	0.28	0.26	0.02–2.88
Divorced	0.82	0.70	0.03–15.48
Widowed	0.72	1.69	0.10–28.95
**Stress**			
Medium	0.47	0.35	0.02–5.94
High	0.96	0.93	0.04–19.88
**Personality**			
Openness	0.13	0.79	0.58–1.07
Neuroticism	0.33	1.12	0.89–1.39
Extroversion	0.93	1.01	0.72–1.42
Responsibility	0.79	1.04	0.78–1.39
Agreeableness	0.11	0.81	0.62–1.05
**Psychological discomfort dimension of the OHIP-49**			
**Group**			
Case	0.03	38.02	1.41–1022.74
**Stress**			
Medium	0.16	8.71	0.41–183.38
High	0.09	16.16	0.66–397.80
**Personality**			
Openness	0.85	1.03	0.78–1.34
Neuroticism	0.04	1.47	1.03–2.11
Extroversion	0.57	0.90	0.65–1.26
Responsibility	0.22	1.18	0.90–1.56
Agreeableness	0.09	1.39	0.95–2.05
**Physical disability dimension of the OHIP-49**			
**Group**			
Case	0.03	45.35	1.50–1367.6
**Stress**			
Medium	0.07	21.65	0.80–588.67
High	0.07	28.30	0.79–1008.4
**Personality**			
Openness	0.31	1.25	0.81–1.94
Neuroticism	0.11	1.28	0.94–1.75
Extroversion	0.81	1.04	0.75–1.45
Responsibility	0.61	0.93	0.70–1.24
Agreeableness	0.91	1.02	0.77–1.33

Reference: Group (0 = control), sex (0 = female), age (continuous), marital status (1 = Single), occupation (1 = Home), stress (categorized 1 = Low, 2 = Medium, 3 = High), personality (continuous). Direct source.

## Data Availability

The data presented in this study are available upon reasonable request to the corresponding author. The data are not publicly available due to privacy restrictions.

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
