# Peer review of "Neuroticism and Psychological Stress in Patients Suffering Oral Lichen Planus: Research Evidence of Psycho-Stomatology?"

_healthcare, 2023, doi:10.3390/healthcare11121738_

Round 1

Reviewer 1 Report

Dear authors,

We have read with interest your manuscript describing an interesting relation between somatic aspects and psychological impacts, in oral dermatologic affections.

The subject is not new, and widely admitted, however your scientific approach based on a case-control study brings reliable insights.

We have some remarks/questions:

1/ Introduction: Could you define and describe more the concepts of Neuroticism, Openness, Agreeableness? These aspects are very important to follow and understand your study.

2/ Discussion: The results are clearly presented, but we miss some interpretations to create a link between clinical prognostic and psychological status

3/ Conclusion: If the questionnaries and methodology are validated, it should lead to a wide clinical investigation, as the limited number of patients included in this study

We believe this study is worth publishing, after these additions-modifications.

Yours faithfully,

Author Response

Dear Reviewer,

First, we would like to thank the reviewers for their time in reviewing our document and for their suggestions and comments. We are sure that they will significantly enhance the manuscript.
In our opinion, we have addressed all the suggestions and comments requested, which we have highlighted in red in the text for better identification of the reviewers.
Looking forward to your comments and instructions, I am pleased to send you my best regards.

Reviewer 1

Q: 1/ Introduction: Could you define and describe more the concepts of Neuroticism, Openness, Agreeableness? These aspects are very important to follow and understand your study.

A: The definition of the concepts of personality traits was included on Page 1, paragraph 1.

Q: 2/ Discussion: The results are clearly presented, but we miss some interpretations to create a link between clinical prognostic and psychological status

A: It was included a sentence on psychological status and clinical prognostic in the Discussion section, Page 9; paragraph 4; and in the Conclusions section, Page 11; paragraph 1

Q: 3/ Conclusion: If the questionnaries and methodology are validated, it should lead to a wide clinical investigation, as the limited number of patients included in this study

A: It was included a sentence on the need to confirm the results with a study involving a large number of patients. in the Conclusions section, Page 11, paragraph 1

For more details, please see the revised manuscript.

Reviewer 2 Report

First of all, I would like to congratulate the authors for your great work. You show in your study that patients suffering from OLP are also associated with higher neuroticism, high stress, less openness and agreeableness and have a poor quality of life.

This is a less investigated topic in the literature and you are helping to shed light on this field. Nevertheless, I have some aspects for you that should be adressed in the manuscript.

1.       Abstract: L.17: OPL should be corrected to OLP

2.       Introduction L.46:   You should explain more about OLP in your introduction (definition/ aetiology/ symptoms/ diagnosis/ regular treatment). Remember that even the non-initiated reader needs to understand what it is all about. Perhaps you can additionally better illustrate the clinical appearance of OLP with a photo of your case group?

3.       Materials and Methods: You write: „Individuals with a diagnosis of premalignant lesion or neoplasia (benign or malignant) were excluded from both groups“. You should explain in the introduction that OLP carries the risk of malignant transformation and support this with relevant literature.

4.       Discussion L.191: We agree that there is little scientific information on this topic. However, in your study you only discuss Čanković et al. Try to include more studies. You could possibly also discuss the following studies:

Mohamadi Hasel K, Besharat MA, Abdolhoseini A, Alaei Nasab S, Niknam S. Relationships of personality factors to perceived stress, depression, and oral lichen planus severity. Int J Behav Med. 2013 Jun;20(2):286-92. doi: 10.1007/s12529-012-9226-5. PMID: 22311191.

Fädler A, Hartmann T, Bernhart T, Monshi B, Rappersberger K, Hof M, Dvorak G. Effect of personality traits on the oral health-related quality of life in patients with oral mucosal disease. Clin Oral Investig. 2015 Jul;19(6):1245-50. doi: 10.1007/s00784-014-1377-0. Epub 2014 Dec 3. PMID: 25467238.

Author Response

Dear Reviewer,

First, we would like to thank the reviewers for their time in reviewing our document and for their suggestions and comments. We are sure that they will significantly enhance the manuscript.
In our opinion, we have addressed all the suggestions and comments requested, which we have highlighted in red in the text for better identification of the reviewers.
Looking forward to your comments and instructions, I am pleased to send you my best regards.

Q: 1.  Abstract: L.17: OPL should be corrected to OLP

A: It was corrected, Page, paragraph 1.

Q: 2. Introduction L.46:   You should explain more about OLP in your introduction (definition/ aetiology/ symptoms/ diagnosis/ regular treatment). Remember that even the non-initiated reader needs to understand what it is all about. Perhaps you can additionally better illustrate the clinical appearance of OLP with a photo of your case group?

A: A paragraph about OLP was included, expanding the clinical-epidemiological and histopathological information, Page 2; paragraph 2. A figure (Figure 1) included clinical examples of reticular and erosive OLP and their histopathological appearance (Page 3).

Q: 3. Materials and Methods: You write: „Individuals with a diagnosis of premalignant lesion or neoplasia (benign or malignant) were excluded from both groups“. You should explain in the introduction that OLP carries the risk of malignant transformation and support this with relevant literature.

A: A paragraph on the OLP as a potentially malignant disease was included in introduction section. Page 2; paragraph 2.

Q: 4. Discussion L.191: We agree that there is little scientific information on this topic. However, in your study you only discuss Čanković et al. Try to include more studies. You could possibly also discuss the following studies:

Mohamadi Hasel K, Besharat MA, Abdolhoseini A, Alaei Nasab S, Niknam S. Relationships of personality factors to perceived stress, depression, and oral lichen planus severity. Int J Behav Med. 2013 Jun;20(2):286-92. doi: 10.1007/s12529-012-9226-5. PMID: 22311191.

Fädler A, Hartmann T, Bernhart T, Monshi B, Rappersberger K, Hof M, Dvorak G. Effect of personality traits on the oral health-related quality of life in patients with oral mucosal disease. Clin Oral Investig. 2015 Jul;19(6):1245-50. doi: 10.1007/s00784-014-1377-0. Epub 2014 Dec 3. PMID: 25467238.

A: The discussion was expanded, and the relation between psychosocial variables and OLP, as well as neuroticism-stress and worse clinical episodes, were discussed. Page 8, paragraph 3; Page 9; paragraphs 1,2,3,4.

References 9, 10, 13, 14, 15, 16, 17, 18, 22 were added. 

For more details, please see the revised manuscript.

Round 2

Reviewer 2 Report

Dear authors, thank you for the changes in the manuscript. For me, there are only minor changes that should be made before publication.

P1, 83: Introduction: It would be nice if you could write something about regular treatments for OLP in the introduction. Precisely because there is no regular cure, your present study on psychological factors is very relevant.

Minor spelling/english errors:

P1, 62: Although its pathogenic mechanism...

P8, 231: Discussion: Exist a well-established association between psychological stresses and oral mucosa diseases has been.
Please check english

P8: 260: OHIP49 should be OHIP-49

P10: 343: despite Therefore, it is suggested...

Author Response

Mexico City, 9th June 9, 2023.

Dear Reviewer (2):

Again, we would like to thank you for taking the time to read the manuscript and make suggestions and comments that we are sure will enrich the article.

In our opinion, we have addressed all the suggestions requested, which we have highlighted in red to facilitate their location.  Looking forward to your comments and instructions, I am pleased to send you my best regards.

 Q: P1, 83: Introduction: It would be nice if you could write something about regular treatments for OLP in the introduction. Precisely because there is no regular cure, your present study on psychological factors is very relevant.

A: A paragraph about the treatment of OLP was included on page 2, paragraph 2.

Minor spelling/english errors:

P1, 62: Although its pathogenic mechanism...

A: The sentence “Although their pathogenic mechanism…” was changed.

P8, 231: Discussion: Exist a well-established association between psychological stresses and oral mucosa diseases has been.
Please check English

A: The sentence was changed: “Exist a well-established association between psychological stresses and some oral mucosa diseases”, page 8, paragraph 2.

Q. P8: 260: OHIP49 should be OHIP-49

A: OHIP49 was changed by OHIP-49, page 8, paragraph 4.  

Q: P10: 343: despite Therefore, it is suggested...

A: “…despite’ was eliminated. Page 10, paragraph 3 (Conclusions).  

Sincerely

The corresponding author.

Dr. Luis Alberto Gaitán Cepeda.

Department of Oral and Maxillofacial Medicine and Pathology, Research and Graduate Division, Dental School, National Autonomous University of Mexico, Mexico City, Mexico. [email protected].